# Myristoyl-CM4 Exhibits Direct Anticancer Activity and Immune Modulation in Hepatocellular Carcinoma: Evidence from In Vitro and Mouse Model Studies

**DOI:** 10.3390/ijms26083829

**Published:** 2025-04-18

**Authors:** Xueli Yuan, Huidan Zhang, Yiqiang Zhu, Ke Xu, Yaxin Yang, Wenjing Xie, Wenliang Duan, Qin Chen, Yuqing Chen

**Affiliations:** Jiangsu Province Key Laboratory for Molecular and Medical Biotechnology, Life Sciences College, Nanjing Normal University, 1# Wenyuan Rd., Nanjing 210000, China; 201202042@njnu.edu.cn (X.Y.); 201201029@njnu.edu.cn (H.Z.); 231202024@njnu.edu.cn (Y.Z.); 21210629@njnu.edu.cn (K.X.); 211202098@njnu.edu.cn (Y.Y.); 221202017@njnu.edu.cn (W.X.); 221202128@njnu.edu.cn (W.D.); 221212025@njnu.edu.cn (Q.C.)

**Keywords:** antimicrobial peptides, myristoyl-CM4, hepatocellular carcinoma, macrophages

## Abstract

Hepatocellular carcinoma (HCC) is a major clinical challenge due to limited treatment options, More therapy candidates with confirmed anticancer effects are urgently needed. Antimicrobial peptide myristoyl-CM4 exhibits effective anticancer activity against leukemia and breast cancer cells. However, its therapeutic potential in HCC remains unexplored. The objective of the present study was to evaluate the anticancer activity of myristoyl-CM4 against cultured HCC cells and HCC xenograft tumors in mice. Cell viability, apoptosis, proliferation, epithelial–mesenchymal transition, migration, and invasion were assessed using standard assays. Mechanistic studies focused on its effects on macrophages utilized western blotting, immunofluorescence staining, and immunohistochemistry assays in HCC/macrophage co-culture models. The study results showed that myristoyl-CM4 induced apoptosis in HCC cells by targeting mitochondria. It also inhibited HCC cell migration and invasion in both HCC monoculture and HCC/macrophage co-culture systems. Notably, myristoyl-CM4 also promoted M1 macrophage polarization and suppressed M2 polarization in co-culture models both in vitro and in vivo. It also demonstrated effective antitumor activity both in PLC-PRF-5 xenograft and PLC-PRF-5/macrophage co-xenograft mouse models. Collectively, these findings highlighted the therapeutic potential of myristoyl-CM4 in HCC treatment.

## 1. Introduction

Liver cancer was the sixth most commonly diagnosed cancer and the third leading cause of cancer death worldwide in 2020, with approximately 906,000 documented new cases and 830,000 deaths [1]. Primary liver cancer includes hepatocellular carcinoma (HCC), comprising 75–85% of cases, and intrahepatic cholangiocarcinoma (10–15%), in addition to other rare types. Chronic hepatic inflammation and liver cirrhosis from any cause, such as chronic viral infections (hepatitis B and C), are the major risk factors for HCC development [2]. Currently, multiple targeted agents, such as tyrosine kinase inhibitors (e.g., sorafenib, lenvatinib) and immunotherapeutics represented by immune checkpoint inhibitors (e.g., nivolumab, pembrolizumab) are included in the treatment guidelines for patients with advanced HCC [3]. However, a large proportion of patients with advanced HCC do not benefit from these systemic therapies due to primary or acquired drug resistance, and HCC remains a highly fatal disease [4]. The high heterogeneity of HCC and its immune-suppressive tumor microenvironment (TME), which aids in immune evasion, represent substantial challenges for effective treatment [3,4,5].

Tumor-associated macrophages (TAMs) are the most abundant immune cells within the HCC TME. They are known to generate an immune-suppressive TME and play a pivotal role in tumor progression [6]. TAMs can be polarized into two main subtypes. The pro-inflammatory/anti-tumorigenic M1 phenotype is induced by factors such as tumor necrosis factor-α, interferon (IFN)-γ, and lipopolysaccharide (LPS), and is characterized by the expression of CD86 and inducible nitric oxide synthase (iNOS), and the anti-inflammatory/tumor-promoting M2-like phenotype is triggered by interleukin (IL)-4 and IL-13, stimulating the and expression of CD163, CD206, and arginase-1 (Arg-1) [7]. In advanced HCC, TAMs predominantly exhibit the M2 phenotype, contributing to tumor progression, invasion, angiogenesis, and immune suppression. As a result, targeting TAMs and reprogramming their M2 phenotype may be a promising therapeutic strategy for HCC [8].

Antimicrobial peptides (AMPs) are small, cationic and amphipathic molecules that play critical roles in host defense against pathogens and are also emerging as important anticancer sources (https://aps.unmc.edu, accessed on 15 February 2025). Several AMPs have shown promising direct anticancer activity against various malignancies, with advantages such as lower drug resistance potential, selective cancer cell targeting of cancer cells, and the ability to prevent primary tumor metastasis by rapidly killing tumor cells [9]. Although AMPs are well known for their anti-bacterial immunity properties, their potential for modulating the TME and enhancing anticancer immunity has recently gained attention, expanding their role in cancer therapy [10,11,12].

Several AMPs derived from humans, insects, animals, and plants, as well as those synthesized artificially, have been investigated to explore their anticancer effects in HCC [13,14,15,16]. However, their efficacy remains limited, highlighting the need for the development of more potent and selective AMPs for HCC treatment. We previously designed a series of fatty-acyl-conjugated AMPs to increase their anticancer activity of AMPs [17]. Among them, myristoyl-CM4 was shown to be effective anticancer activity against leukemia and breast cancer cells, both in vitro and in vivo, while also exhibiting low toxicity towards normal cells [18,19]. However, its therapeutic potential in HCC remains unexplored. The present study investigated the anticancer effects of myristoyl-CM4 on HCC cells. The study findings demonstrated that myristoyl-CM4 inhibited HCC cell growth via apoptosis induction, migration inhibition, and epithelial–mesenchymal transition (EMT) suppression. Moreover, it enhanced M1 macrophage polarization in an HCC/macrophage co-culture system, both in vitro and in vivo. These results suggested that myristoyl-CM4 could be a potential therapeutic agent for HCC treatment.

## 2. Results

### 2.1. Myristoyl-CM4 Induces Cytotoxicity in HCC Cells

The cytotoxicity of myristoyl-CM4 against HCC cells was evaluated using the CCK-8 assay (Figure 1A). A dose-dependent cytotoxic effect was observed following myristoyl-CM4 treatment. The IC50 of myristoyl-CM4 against PLC/PRF-5 and HepG2 cells was ~4 µM. This value was ~18 µM against a normal human hepatic stellate cell line LX-2. These results demonstrated that myristoyl-CM4 exhibited a higher selectivity for HCC cells over normal hepatic cells. Furthermore, minimal lactate dehydrogenase (LDH) release was detected from PLC/PRF-5 cells at 4 µM of myristoyl-CM4, whereas 20 µM myristoyl-CM4 led to a 40% LDH release (Figure 1B). Flow cytometry analysis revealed a significant increase in the fluorescence intensity of FITC-labeled myristoyl-CM4, suggesting a strong binding affinity to the PLC/PRF-5 cell surface (Figure 1C). Co-localization of myristoyl-CM4 with mitochondria was observed in both PLC/PRF-5 and HepG2 cells, indicating its ability to penetrate cells and bind to mitochondria (Figure 1D). Western blot analysis showed an increase in cleaved caspase-9 and caspase-3, and a decrease in Bcl-2 and PARP levels, suggesting that myristoyl-CM4 induced mitochondria-dependent apoptosis in both PLC/PRF-5 and HepG2 cells (Figure 1E). Together, these results suggested that myristoyl-CM4 exhibited its anticancer activity partly by inducing mitochondria-dependent apoptosis.

### 2.2. Myristoyl-CM4 Inhibits Migration and Invasion of Cultured HCC Cells

Wound healing and transwell invasion assays were performed to determine the effect of myristoyl-CM4 on HCC cell migration and invasion. Treatment with myristoyl-CM4 (1 µM, 2 µM) significantly inhibited the migration (*p* < 0.001; Figure 2A) and reduced the invasion (*p* < 0.001; Figure 2B) of PLC/PRF-5 and HepG2 cells. Additionally, myristoyl-CM4 (2 µM) significantly decreased the expression of mesenchymal markers (N-cadherin and vimentin) in both PLC/PRF-5 and HepG2 cells (*p* < 0.001; Figure 2C). The level of EMT-associated transcription factor Snail was also significantly reduced after the myristoyl-CM4 treatment. These results demonstrated that myristoyl-CM4 directly inhibited the migration and invasion of cultured HCC cells.

### 2.3. Myristoyl-CM4 Inhibits the Growth of PLC/PRF-5 Xenograft Tumor Growth

A PLC/PRF-5 xenograft mouse was established by subcutaneously injecting PLC/PRF-5 cells into BALB/c nude mice in order to evaluate the in vivo antitumor effects of myristoyl-CM4. No significant differences in body weight were observed between the myristoyl-CM4 (5 mg/kg) and phosphate-buffered saline (PBS) control groups (Figure 3A). Tumor growth curves revealed that myristoyl-CM4 significantly inhibited tumor growth. Tumor volume in the myristoyl-CM4 group had decreased more than four-fold (tumor volume: 130.48 ± 12.59 mm^3^ vs. 566.75 ± 57.87 mm^3^) compared to that in the control group on the 30th day (Figure 3B,C). The tumor inhibition rate in the myristoyl-CM4 group reached 70% compared to that in the control group (Figure 3D).

Immunofluorescence staining for PCNA and Ki67 indicated a significant reduction in the number of proliferating cells in the myristoyl-CM4-treated group (Figure 3E). Histological analysis of tumor tissues using an anti-CD31 antibody, a marker of vascularization, showed decreased numbers of CD31+ cells in the myristoyl-CM4-treated tumors. Western blot analysis further revealed that myristoyl-CM4 treatment decreased cyclin D1 and PCNA levels, and increased cytochrome c, Bak, cleaved caspase-9, and cleaved PARP levels, indicating obvious induction of mitochondria-dependent apoptosis by myristoyl-CM4 treatment (Figure 3F). These results suggested that myristoyl-CM4 exhibited effective antitumor activity in the PLC/PRF-5 xenograft model by inducing apoptosis, inhibiting proliferation, and reducing vascularization.

### 2.4. Myristoyl-CM4 Inhibits HCC Cell Survival in the HCC/Macrophage Co-Culture System

To assess the cytotoxicity of myristoyl-CM4 in macrophages, THP-1-derived M0 macrophages were induced with PMA, followed by M1 and M2 polarization using LPS/IFN-γ and IL-4, respectively. M1 and M2 polarization status was confirmed by examining classical markers (iNOS, CD86 for M1; CD163 for M2) (Figure 4A–D). Myristoyl-CM4 (1 µM) exhibited low cytotoxicity (cell viability > 90%) to M0, M1, and M2 macrophages (Figure 4E). A concentration of ≤1 µM was used in subsequent studies. Co-culture of PLC/PRF-5 cells with M0 macrophages for 48 h promoted HCC cell growth, while myristoyl-CM4 (0.5 µM, 1 µM) significantly inhibited the growth of PLC/PRF-5 cells in the co-culture system (Figure 4F). These findings suggested that myristoyl-CM4 may inhibit HCC cell growth by modulating macrophage activity.

### 2.5. Myristoyl-CM4 Inhibits Proliferation, EMT, Migration, and Invasion of HCC Cells in the HCC/M0 Co-Culture System

The effects of myristoyl-CM4 (0.5 µM, 1 µM) on HCC cell proliferation, migration, invasion, and EMT of HCC cells in the HCC/macrophage co-culture system were further investigated. Conditioned medium (CM) from myristoyl-CM4-treated M0 macrophages inhibited the proliferation of PLC/PRF-5 and HepG2 cells, resulted from a reduction in the number of EdU-positive cells (*p* < 0.001; Figure 5A). Wound healing and transwell invasion assays showed significant inhibition of migration and invasion when HCC cells were cultured in myristoyl-CM4-treated CM (*p* < 0.01; Figure 5B,C). Furthermore, western blotting revealed that myristoyl-CM4 treatment significantly increased E-cadherin expression while decreasing N-cadherin and vimentin levels, indicating EMT suppression by myristoyl-CM4 (*p* < 0.001; Figure 5D). These results suggested that myristoyl-CM4-treated M0 macrophages effectively inhibited proliferation, migration, invasion, and EMT of HCC cells.

### 2.6. Myristoyl-CM4 Promotes Macrophage M1 Polarization

A co-culture system was established using 24-well transwell plates to explore the impact of myristoyl-CM4 on macrophage polarization. A co-culture system was established using 24-well transwell plates with M0 macrophages seeded in the bottom compartment and PLC/PRF-5 or HepG2 cells in the upper compartment. Macrophages were then collected to analyze the polarization markers after 48 h of myristoyl-CM4 treatment (0.5 µM, 1 µM). Results showed that myristoyl-CM4 significantly increased the levels of M1 markers (iNOS, CD86) (Figure 6A). Immunofluorescence staining confirmed these findings, showing increased iNOS and decreased CD163 levels in myristoyl-CM4-treated macrophages (Figure 6B). Flow cytometry analysis also demonstrated an increase in CD86-FITC fluorescence (Figure 6C). These results suggested that myristoyl-CM4 (0.5 μM, 1 μM) promotes macrophages towards a M1 phenotype.

### 2.7. Myristoyl-CM4 Regulates Macrophage Polarization and Inhibits Tumor Growth in Co-Xenografts Tumor Models

An HCC/macrophage co-xenograft mouse model was established by subcutaneously injecting PLC/PRF-5 cells with M0 macrophages into nude mice in order to evaluate the in vivo anti-tumor activity of myristoyl-CM4 in the presence of TAMs. The mice were treated with PBS, myristoyl-CM4 (2.5 mg/kg, 5 mg/kg), or lenvatinib (5 mg/kg) when the tumors reached a volume of 75–100 mm^3^. Results showed no significant differences in body weight among groups with different treatments (Appendix A). Tumor growth was significantly promoted by macrophages in the co-xenograft model compared to that in the PLC/PFR-5 xenograft model (tumor volume: 1, 1270.03 ± 150.05 mm^3^ vs. 566.75 ± 57.87 mm^3^), while myristoyl-CM4 treatment inhibited co-xenograft tumor growth (Figure 7A,B). The tumor inhibition rates for 2.5 mg/kg and 5 mg/kg myristoyl-CM4 treatments were 48% and 65%, respectively, which were comparable to those after the lenvatinib treatment (inhibition rate: 62%), a first-line therapy for patients with advanced HCC (Figure 7C). Decreased N-cadherin, Snail, and vimentin levels, and increased E-cadherin expression level were detected in myristoyl-CM4-treated tumors (Figure 7D). Western blotting outcomes revealed significantly increased M1 markers (iNOS, CD86) and decreased M2 marker (CD163, Arg-1) levels in myristoyl-CM4-treated (2.5 mg/kg) tumors (*p* < 0.001, Figure 7E). Immunohistochemistry results also confirmed higher iNOS and lower CD163 levels in the myristoyl-CM4-treated groups (Figure 7F). These findings suggested that myristoyl-CM4 promoted macrophages towards a M1 phenotype, and inhibited PLC/PRF-5/macrophage co-xenograft tumor growth in mouse models.

## 3. Discussion

Current HCC treatment strategies for hepatocellular carcinoma (HCC) are limited, and there is an urgent need for new therapeutic candidates with confirmed anticancer effects in order to develop more effective drugs for HCC treatment. Among the potential candidates, AMPs derived from naturally occurring or modified peptides have garnered significant attention due to their multiple advantages over current treatment modalities [20]. The present in vitro and in vivo mouse model studies demonstrated that myristoyl-CM4 exhibited effective anticancer activity against HCC through the following dual mechanisms: direct effects on HCC cells and immunoregulatory modulation via macrophage polarization.

Previous studies showed that myristoyl conjugation significantly enhances the anticancer activity of CM4 against breast cancer (MX-1, MCF-7, and MDA-MB-231) and leukemia cells (K562/MDR, Jurkat) cells in vitro, with IC50 values ranging from 2 µM to 6 µM [17,18,19]. The present study findings further validated the effective anticancer activity of myristoyl-CM4 against HCC cells (PLC/PRF-5 and HepG2), with an IC50 of ~4 µM. Comparative analysis of IC50 among all reported AMPs against HCC cells revealed that myristoyl-CM4 demonstrated superior anticancer potency to most of the reported AMPs, with melittin and its derivatives being the only compound that surpassed it in efficacy. Notably, melittin and its derivatives demonstrate broad-spectrum cytotoxicity against both cancerous cells and normal cells [21]. Generally, the anticancer mechanisms of AMPs can be categorized into two primary types: selective membrane disruption and non-membranolytic actions, with apoptosis being a common feature of most non-membranolytic actions [22,23]. Similar to our previous studies in leukemia and breast cancer cells, myristoyl-CM4 can be internalized by HCC cells, inducing mitochondria-dependent apoptosis. Thus, it seemed that (1) myristoyl-CM4 exhibited broad-spectrum anticancer activity against several tumor types, maintaining an IC50 in the low micromolar range; (2) its common mechanism of action involved internalization, mitochondria targeting, and activating mitochondria-dependent apoptosis across different cancer cell types; and (3) myristoyl-CM4 exhibited lower cytotoxicity in normal cells. Thus, myristoyl-CM4 emerged as one of the most promising AMPs for anticancer candidates.

Therapeutic options for HCC are still limited at present, and metastasis remains a key factor contributing to the high recurrence rate and poor prognosis associated with this disease [24,25]. EMT enables tumor cells to acquire highly malignant traits and plays a crucial role in HCC metastasis [26]. Several biomarkers, including N-cadherin, E-cadherin, and Vimentin, have been implicated in EMT [27]. A few AMPs, such as melittin, cathelicidin, and human β-defensins, have been reported to inhibit the migration of monocultured cancer cells [28,29,30]. In the present study, myristoyl-CM4 effectively inhibited EMT, migration, and invasion both in monocultured HCC cells and an HCC/macrophage co-culture system at concentrations below the IC50 level. The mechanism of action likely varies with peptide concentration. At lower concentrations, myristoyl-CM4 may bind to the cell membrane to initiate inhibitory signals for migration. At higher concentrations (near the IC50), it could internalize into cells, targeting intracellular components such as mitochondria, to induce apoptosis. This concentration-dependent AMP behavior has been noted in several studies [31,32]. These additional anticancer activities further emphasized the potential of myristoyl-CM4 as a promising candidate for HCC treatment.

TAMs are the most abundant immune cells in the HCC microenvironment that can be polarized into two main subsets of M1 and M2 depending on TME signals [33]. M1 macrophages are involved in tumor suppression, while M2 macrophages promote tumor growth. Indeed, the present in vivo results demonstrated a large number of CD163 positive cells in tumor tissues of co-xenografts mice. Furthermore, co-xenografts (PLC/PRF-5 cells with macrophages) led to a more rapid tumor growth compared to the PLC/PRF-5 xenografts alone, highlighting the pro-tumor role of macrophages in the TME. Previous studies have shown that HCC cells secrete various molecules, such as IL-4, IL-13, CSF-1, CCL2, and CXCL12, that trigger M2 macrophage polarization [34]. Given the pivotal role of TAMs in supporting tumor progression, targeting them has become an attractive strategy for cancer therapy [35]. However, there are few studies on their effect on macrophages in TME, and reports mainly focus on melittin and its derivatives [10,11,12]. These peptides showed the activity to eliminate the M2-like TAMs. The present study revealed a novel anticancer AMP (myristoyl-CM4) that acted on macrophages. More interestingly, myristoyl-CM4 (0.5 µM, 1 µM) could significantly increase M1 marker levels (iNOS, CD86) in the HCC/macrophage co-culture system, especially in PLC/PRF-5/macrophage co-xenograft tumors, demonstrating its ability to promote macrophage M1 polarization. This suggested that myristoyl-CM4’s antitumor effect may, in part, result from its modulation of the macrophage polarization towards a more tumor-suppressive M1 phenotype, which further expands the anticancer mechanism and application prospects of AMPs.

Several AMPs have been reported to exhibit their anti-inflammatory properties by driving macrophage polarization from M1 to M2 phenotypes in inflammatory contexts, such as pathogen infection and LPS induction [36,37]. Interestingly, low-dose administration of myristoyl-CM4 in the HCC/macrophage co-culture system induces M0 macrophage polarization toward the M1 phenotype, revealing a proinflammatory response within the HCC microenvironment. Further mechanistic studies elucidating the crosstalk between myristoyl-CM4 and TME components in HCC tumors are needed. Additionally, apoptosis induction is an important mechanism, and the detailed interaction between myristoyl-CM4 and mitochondria, and mitochondrial proteins remains to be further explored. In order to develop the clinical application of myristoyl-CM4, its stability in physiological conditions, the long-term effects, and pharmacokinetics also need to be investigated in the future.

## 4. Materials and Methods

### 4.1. Reagents

Antibodies against Bcl-2, caspase-3, caspase-9, PCNA, CD31 and PARP were purchased from Cell Signaling Technology (Danvers, MA, USA). Antibodies against E-cadherin, N-cadherin, Snail, Vimentin, iNOS, Arg-1, CD163, CD68 and Cyclin D1 were purchased from Santa Cruz Biotechnology Inc. (Santa Cruz, CA, USA). Antibodies against β-actin, anti-rabbit secondary antibody and anti-mouse secondary antibody were purchased from Abclonal (Wuhan, China). The myristoyl-CM4 and FITC-myristoyl-CM4 were synthesized by Synpeptide Inc. (Nanjing, China). Lenvatinib was purchased from MCE (Princeton, NJ, USA). RPMI-1640, DMEM, IL-4, LPS and IFN-γ were purchased from Thermo Fisher Scientific (Waltham, MA, USA), and fetal bovine serum (FBS) from Capricorn Scientific (Hessen, Germany). CCK-8, Trizol reagent, Taq Master Mix, HiScriptIII RT SuperMix for qPCR (+gDNA wiper), AceQ qPCR SYBR Green Master Mix (High ROX Premixed) and BCA Protein Assay were purchased from Vazyme (Nanjing, China). The MTT Assay Kit was purchased from Keygen Biotech (Nanjing, China). DAB and DAPI were purchased from Solarbio (Beijing, China). PMA, LDH cytotoxicity test kits, and the BeyoClick™ EdU-555 kit were purchased from Beyotime (Nanjing, China).

### 4.2. Cell Culture

Human HCC cell lines PLC/PRF-5, HepG2, human normal hepatic stellate cell line LX-2, and human acute monocytic leukemia-cell line THP-1 were obtained from the American Type Culture Collection (ATCC, Shanghai, China). PLC/PRF-5, HepG2 and LX-2 cells were cultured in DMEM medium supplemented with 10% fetal bovine serum (FBS), 100 U/mL penicillin and 100 μg/mL streptomycin. THP-1 cells were maintained in RPMI 1640 medium containing 10% FBS, 100 U/mL penicillin and 100 μg/mL streptomycin. For the preparation of conditioned medium (CM) from macrophages, the culture media of macrophages were collected and centrifuged at 12,000 rpm for 5 min. The supernatants were filtered using 0.22 μm pore size membranes and stored at −20 °C for up to one month until further use.

### 4.3. Macrophage Differentiation and Polarization

THP-1 cells were differentiated into M0 macrophages by stimulation with 100 ng/mL phorbol 12-myristate 13-acetate (PMA) for 24 h. For M1 macrophage polarization, 100 ng/mL LPS and 20 ng/mL IFN-γ were added to M0 macrophages for 48 h. For M2 macrophage polarization, 20 ng/mL IL-4 was added to M0 macrophages for 48 h.

### 4.4. CCK-8 Assay

The CCK-8 assay was performed according to the manufacturer’s instructions. Briefly, cells were seeded in 96-well plates at a density of 1 × 10^4^ cells/well and treated with different concentrations of peptides for 24 h. Then, CCK-8 solution was added to each well and incubated at 37 °C for 2 h. Absorbance was measured at 450 nm using a Synergy H1 Multifunction Microplate Reader (Synergy, Perth, Australia).

### 4.5. MTT Assay

THP-1, PLC/PRF-5, and HepG2 cells were seeded in 96-well plates at a density of 1 × 10^4^ cells/well and treated with different concentrations of peptides for 24 h. Then, 50 μL of MTT solution was added to each well, and cells were incubated at 37 °C for 4 h. The reaction was stopped by adding 150 μL of DMSO to dissolve the formazan. Absorbance at 490 nm was measured using a Synergy H1 Multifunction Microplate Reader.

### 4.6. Lactate Dehydrogenase (LDH) Release Assay

PLC/PRF-5 cells (1 × 10^5^/mL) were incubated with different concentrations of myristoyl-CM4 for different times. PBS and 2% Triton X-100 were used as controls. After centrifugation, 100 μL supernatant was transferred and a tetrazolium salt reaction mixture was added. The absorbance of the colored formazan was determined using a microplate reader at 490 nm. LDH leakage (%) was calculated using the following equation: [(A − C)/(P − C)] × 100, where A is the mean absorbance in the test wells, C is the PBS control, and P is the Triton X-100 control. Data were reported as mean ± SEM of four independent experiments

### 4.7. Cell Binding Assay and Rh123 Staining

PLC/PRF-5 and HepG2 cells were incubated with FITC-myristoyl-CM4 at 37 °C for 5, 20, or 30 min in the dark. After washing with PBS, the cells were analyzed by FACS Vantage SE flow cytometer. Some cells were subjected to Rhodamine123 (Rh123) staining. After fixation with 4% paraformaldehyde, cells were incubated with Rh123 and FITC-myristoyl-CM4 at 37 °C for 1 h. Cells were then washed with PBS, and fluorescence microscopy was performed using a Leica DMi8 fluorescence microscope.

### 4.8. Western Blotting

Cells were lysed using RIPA buffer containing protease and phosphatase inhibitors, and protein concentrations were determined using a BCA Protein Assay Kit. Proteins (20 μg) were separated by SDS-PAGE and transferred to PVDF membranes (Millipore, Darmstadt, Germany), which were blocked with 5% skim milk for 1 h. Membranes were incubated with primary antibodies at 4 °C overnight. After washing, membranes were incubated with HRP-conjugated anti-mouse secondary antibodies for 1 h. Blots were detected using an ECL substrate and imaged with a Chemiluminescent Imaging System (Tanon, Shanghai, China). Image analysis was performed using ImageJ (version 1.54 d) software.

### 4.9. Quantitative Real-Time PCR (qRT-PCR)

Total RNA was extracted using Trizol reagent. cDNA was synthesized using HiScriptIII RT SuperMix for qPCR (+gDNA wiper). The qPCR was performed using AceQ qPCR SYBR Green Master Mix, and relative mRNA expression levels were normalized to β-actin. The 2^−ΔΔCt^ method was used for quantification. Primer sequences are provided in Appendix A.

### 4.10. Wound Healing Assay

PLC/PRF-5 and HepG2 cells (2 × 10^5^) were seeded in 12-well plates until cell adhesion. A cell-free area was created by scraping with a 200 μL pipette tip, and DMEM or conditioned medium from macrophages containing 1% FBS was added. Migration rates were observed at 0, 24, and 48 h using a microscope, and the results were analyzed using ImageJ software.

### 4.11. Transwell Invasion Assay

Transwell invasion assays were conducted to assess cell invasive capacity. Transwell chambers with 8-μm pore size (Corning, NY, USA) were coated with 100 μL of 1:8 diluted Matrigel (BD Biosciences, San Jose, CA, USA) and incubated at 37 °C for 4 h. 5 × 10^4^ PLC/PRF-5 or HepG2 cells in 200 μL DMEM were seeded in the upper chamber of transwell inserts (Corning, NY, USA), with the lower chamber containing 500 μL of DMEM supplemented with 15% FBS or conditioned medium (CM). After 24 h of incubation, cells that migrated to the lower chamber were fixed with 4% paraformaldehyde, stained with 0.1% crystal violet, and imaged using a light microscope.

### 4.12. EdU Incorporation Assay

The EdU assay was performed using the BeyoClick™ EdU-555 kit according to the manufacturer’s instructions. Briefly, cells were treated with EdU (20 μM) for 2 h at 37 °C. After fixation with 4% paraformaldehyde and permeabilization with 1% Triton X-100, cell nuclei were stained with DAPI for 15 min in the dark. Imaging was performed using a Ti-E-A1R confocal laser microscope (Nikon, Tokyo, Japan).

### 4.13. Immunofluorescent Staining

THP-1 cells were plated on 24-well glass slides and polarized to M0/M1/M2 macrophages. After treatment, cells were fixed with 4% paraformaldehyde and permeabilized with 1% Triton X-100. Cells were incubated with primary antibodies against Arg-1, CD163, or iNOS at 4 °C overnight. After washing, cells were incubated with secondary antibodies conjugated with fluorescent dyes for 1 h in the dark. DAPI was used for nuclear counterstaining. Fluorescence imaging was performed using a fluorescence microscope (Nikon, Tokyo, Japan).

### 4.14. Xenograft Tumor Model in Nude Mice

Male Balb/c nude mice (4–6 weeks old, 18.0 ± 2.0 g) were purchased from GemPharmatech Co., Ltd. (Nanjing, China) and housed in specific pathogen-free (SPF) facilities. All animal protocols were approved by the Ethics Committee of Nanjing Normal University (grant number IACUC-20220218). To establish the tumor model, 5 × 10^6^ PLC/PRF-5 cells or a mixture of 5 × 10^6^ PLC/PRF-5 cells and 1.5 × 10^6^ M0 macrophages were injected subcutaneously. Mice were randomly divided into six groups (4–6 mice per group) including (1) PBS group (PLC/PRF-5 cells), (2) 5 mg/kg myristoyl-CM4 group, (3) PBS group (PLC/PRF-5 cells and M0 macrophages), (4) 2.5 mg/kg myristoyl-CM4 group (PLC/PRF-5 cells and M0 macrophages), (5) 5 mg/kg myristoyl-CM4 group (PLC/PRF-5 cells and M0 macrophages), and (6) 5 mg/kg Lenvatinib group (PLC/PRF-5 cells and M0 macrophages). Tumor volume and weight of each mouse were recorded. Tumor volume was calculated as length × width^2^/2. After 4 weeks, the tumor tissues were dissected from euthanized mice, imaged and analyzed. The rate of tumor growth inhibition was calculated from the tumor weight (TW, g) according to the following formula: tumor growth inhibition rate = (TW_control_ − TW_treatment_)/TW_control_ × 100%.

### 4.15. Immunohistochemistry (IHC)

Tissues were first cut into small pieces, fixed and cryoprotected in 4% paraformaldehyde for 24 h. Frozen samples were then sectioned into pieces at 6 μm thickness. Sectioned specimens were attached onto slides, washed and incubated with 3% H_2_O_2_ in methanol for 10 min to quench endogenous peroxidase. After washing, the sections were blocked and probed with the indicated primary antibodies at 4 °C overnight, including rabbit anti-PCNA, anti-CD31, anti-CD163, anti-iNOS and anti-vimentin antibodies. After washing, the sections were probed with goat anti-rabbit IgG-HRP for 1 h at room temperature, followed by incubation with DAB substrate for signal development. After incubation with hematoxylin for counterstain, the sections were subjected to a Ti-E-A1R confocal laser microscope for imaging. Approximately 4–6 fields were randomly selected for each sample. Data were collected from five independent experiments.

### 4.16. Statistical Analysis

All experiments were performed at least three times, with one representative experiment shown. The data are represented as means ± SEM. Two-tailed Student’s *t*-test and one-way ANOVA followed by Tukey’s multiple comparison test were used to determine the significance of differences. For all cases, *p* < 0.05 was considered statistically significant. Statistical analysis was assessed using GraphPad Prism 6. Statistical significance was determined as indicated in the figure legends. The symbols *, **, and *** denote significance at the 0.05, 0.01, and 0.001 levels, respectively.

## 5. Conclusions

The present in vitro and in vivo mouse model studies demonstrate that myristoyl-CM4 exhibits effective anticancer activity against HCC via direct effects and immune modulation. These direct effects on HCC cells include the induction of apoptosis and the inhibition of EMT, migration, and invasion. Additionally, myristoyl-CM4 modulates the immune microenvironment by promoting macrophage M1 polarization, thereby contributing to anti-tumor immunity. Collectively, its effective antitumor activity and multiple mechanisms of anticancer action highlight the therapeutic potential of myristoyl-CM4 in HCC treatment.

## Figures and Tables

**Figure 1 ijms-26-03829-f001:**
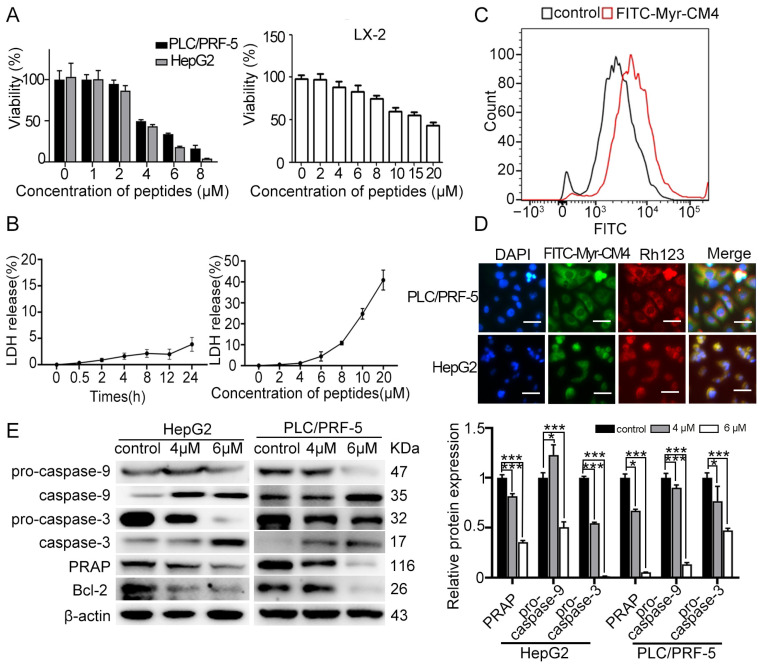
Effect of myristoyl-CM4 on the cytotoxicity of cultured HCC cells. (**A**) CCK-8 assay and (**B**) LDH release assays were conducted in PLC/PRF-5, HepG2, and LX-2 cells after treatment with different concentrations of myristoyl-CM4. (**C**) Binding affinity of myristoyl-CM4 to PLC/PRF-5 cells. Cells were incubated with 3 μM FITC-labeled peptides for 30 min at 37 °C and then analyzed by flow cytometry. (**D**) Co-localization analysis of myristoyl-CM4 with mitochondria in PLC/PRF-5 and HepG2 cells. “—” represents the scale bar of 50 μm. (**E**) Western blot analysis of apoptotic proteins (caspase-9, caspase-3, Bcl-2, and PARP). Relative amounts of procaspase-9, procaspase-3, and PARP vs. β-actin were determined based on western blotting results and ImageJ (version 1.54 d) densitometry analysis. Data are presented as mean ± SEM from 3 independent experiments. * *p* < 0.05, *** *p* < 0.001.

**Figure 2 ijms-26-03829-f002:**
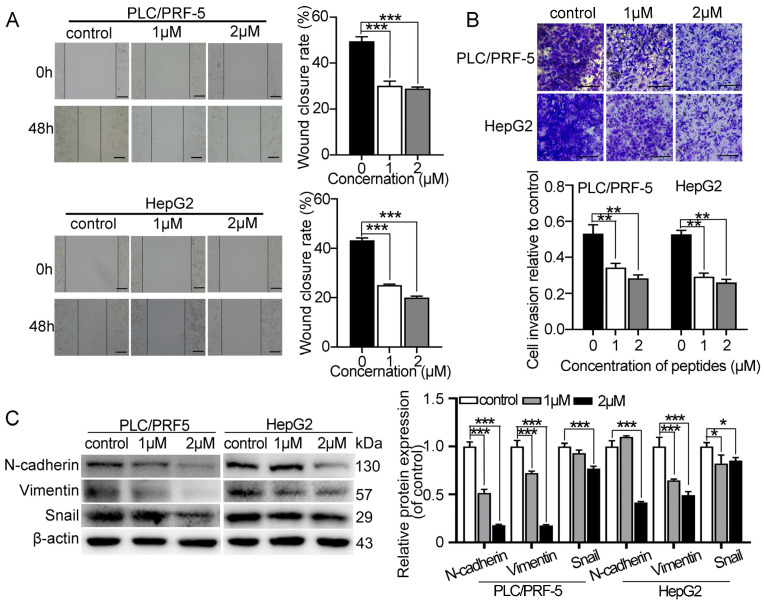
Effect of myristoyl-CM4 on the migration, invasion and EMT in HCC cells. PLC/PRF-5 and HepG2 cells were treated with different concentrations (1 μM, 2 μM) of myristoyl-CM4. HCC cell migration and invasion were detected by (**A**) wound healing assay and (**B**) transwell invasion assays, respectively. Representative wound healing assay images were taken at 0 h and 48 h. Transwell assay images were acquired at 24 h. Wound closure rate (%) was calculated based on three independent experiments. “—” represents scale bar of 100 μm. (**C**) Protein levels of N-cadherin, vimentin, and Snail were detected by western blotting. Data are represented as mean ± SEM of 3 independent experiments. * *p* < 0.05, ** *p* < 0.01, *** *p* < 0.001.

**Figure 3 ijms-26-03829-f003:**
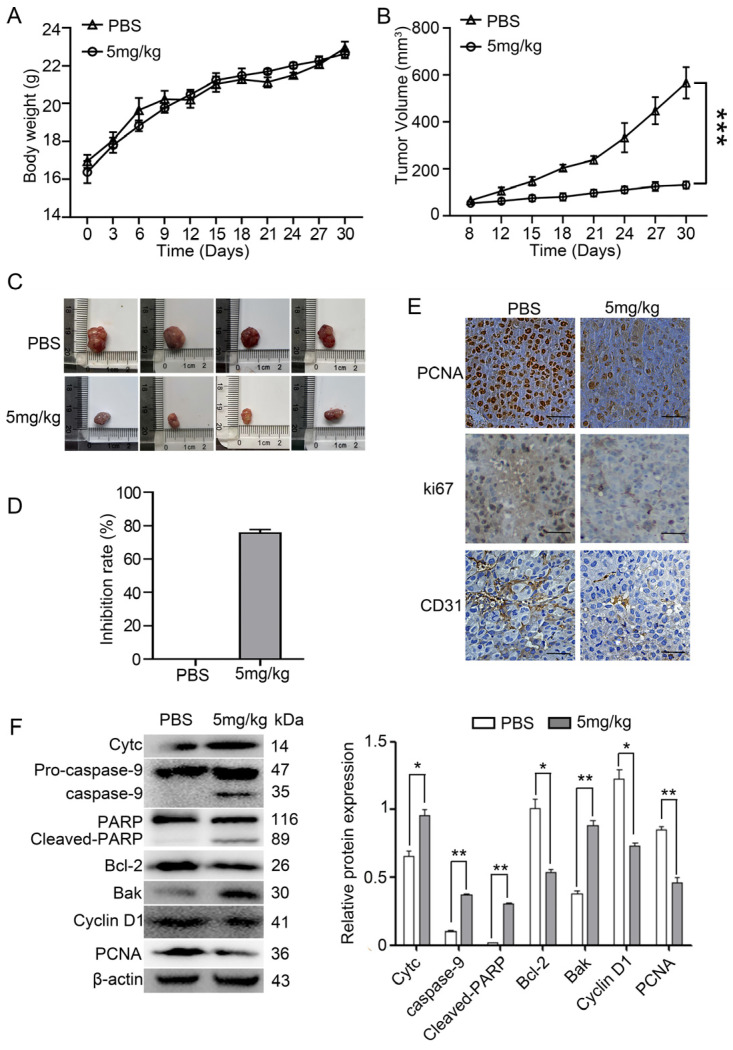
Antitumor effect of myristoyl-CM4 in PLC/PRF-5 xenograft mouse model. A PLC/PRF-5 xenograft model was established by subcutaneously injecting PLC/PRF-5 cells into BALB/c nude mice aged 4–6 weeks. Mice were divided into two groups; one was injected with PBS and the other injected with 5 mg/kg myristoyl-CM4. (**A**) Body weight and (**B**,**C**) tumor volume were determined every three days. Tumor growth curves over time were generated until mice were sacrificed on day 30. (**D**) Tumor inhibition rate was calculated according to tumor weight. (**E**) PCNA, CD31 and Ki67 expression levels were determined by immunohistochemistry. Scale bar represents 100 μm. (**F**) Lysates from tumor tissue were harvested and the levels of cytochrome c, Bak, caspase-9, caspase-3, PARP, cyclin D1, and PCNA levels were determined using western blotting. ns, no significance * *p* < 0.05, ** *p* < 0.01, *** *p* < 0.001.

**Figure 4 ijms-26-03829-f004:**
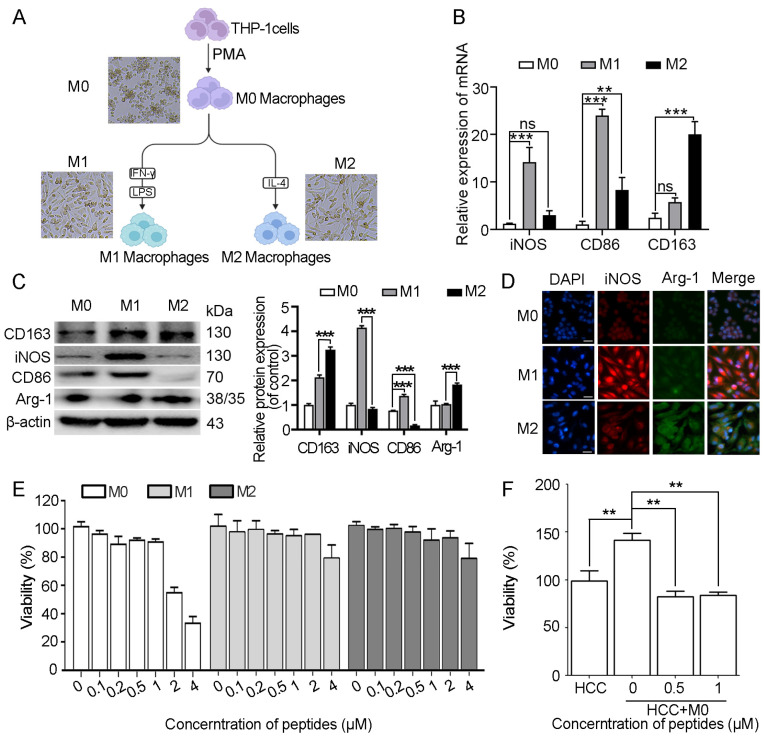
Effect of myristoyl-CM4 on macrophages and HCC cells in co-culture system. (**A**) Schematic illustration of differentiation and polarization of THP-1-derived macrophages. Morphological characteristics of macrophages after polarization were observed under microscope. Scale bar represents 50 μm. (**B**) Real-time PCR and (**C**) western blot assays were used to detect the expression of macrophage marker proteins at protein level and mRNA levels. “—” represents scale bar of 50 μm. (**D**) Immunofluorescence analysis was conducted to assess the levels of iNOS and Arg-1 levels. (**E**) Cytotoxicity of myristoyl-CM4 on M0/M1/M2 macrophages was determined by MTT assay. (**F**) M0 macrophages were placed on the upper layer, while PLC/PRF-5 cells were placed on the bottom layer. MTT assay was conducted on PLC/PRF-5 cells after 48 h of co-culture. ns, no significance. ** *p* < 0.01, *** *p* < 0.001.

**Figure 5 ijms-26-03829-f005:**
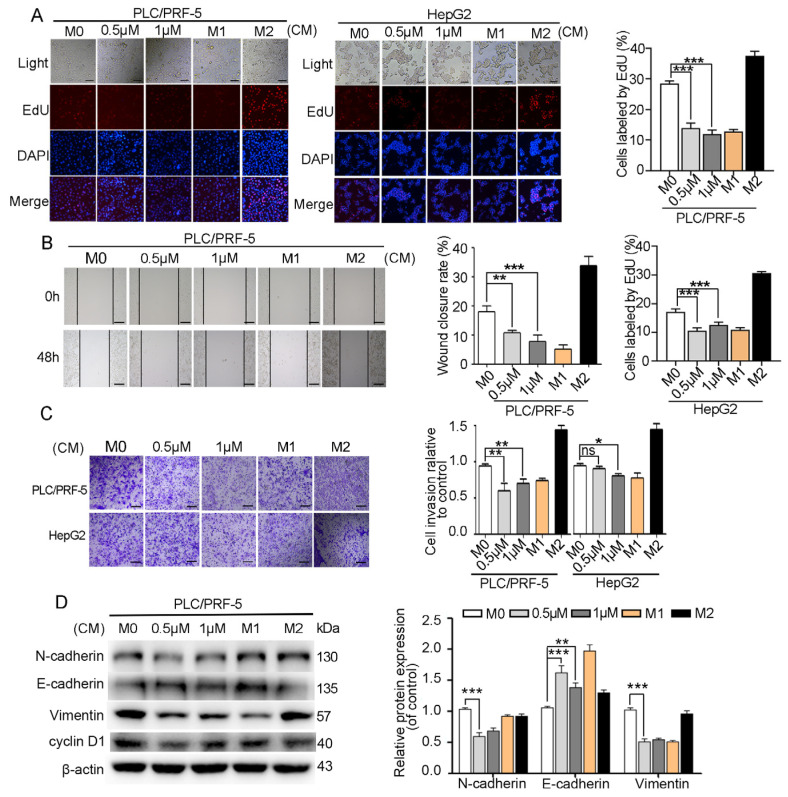
Effect of myristoyl-CM4 on HCC cells in HCC/M0 co-culture system. M0 macrophages were treated with myristoyl-CM4 for 48 h. Conditioned medium (CM) was then collected to culture HCC cells for 48 h. (**A**) EdU analysis of PLC/PRF-5 and HepG2 cell proliferation. (**B**) Wound healing and (**C**) transwell invasion assays in PLC/PRF-5 and HepG2 cells were conducted. Representative wound healing assay images were taken at 0 h and 48 h. Transwell assay images were acquired at 24 h. Wound closure (%) was calculated based on four independent experiments, and expressed as mean ± SEM. (**D**) EMT markers were detected using western blotting. “—” represents the scale bar of 100 μm. ns, no significance. * *p* < 0.05, ** *p* < 0.01, *** *p* < 0.001.

**Figure 6 ijms-26-03829-f006:**
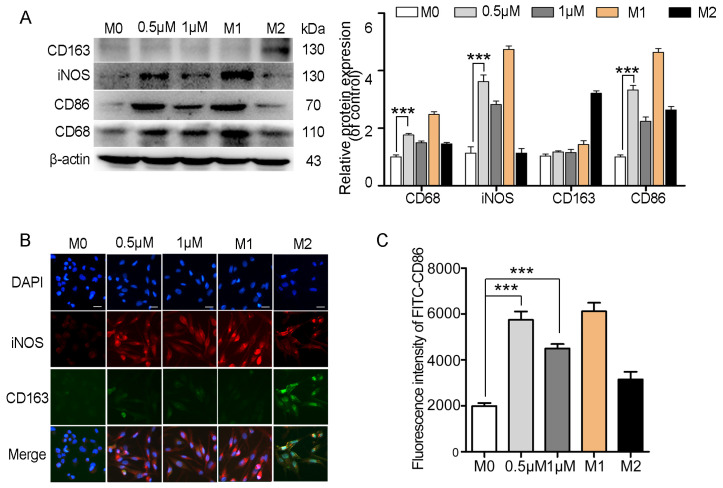
Myristoyl-CM4 promotes macrophage M1 polarization. M0 macrophages were seeded in the bottom compartment, while PLC/PRF-5 cells were cultured in the upper compartment. Macrophages were collected after myristoyl-CM4 (0.5 μM, 1 μM) treated for 48 h. (**A**) Macrophages lysates were collected and the levels of iNOS, CD86 and CD163 levels were determined using western blotting. (**B**) Immunofluorescence staining for iNOS and CD163 in macrophages. “—” represents scale bar of 50 μm. (**C**) FITC-CD86 fluorescence intensity was detected using flow cytometry analysis. *** *p* < 0.001.

**Figure 7 ijms-26-03829-f007:**
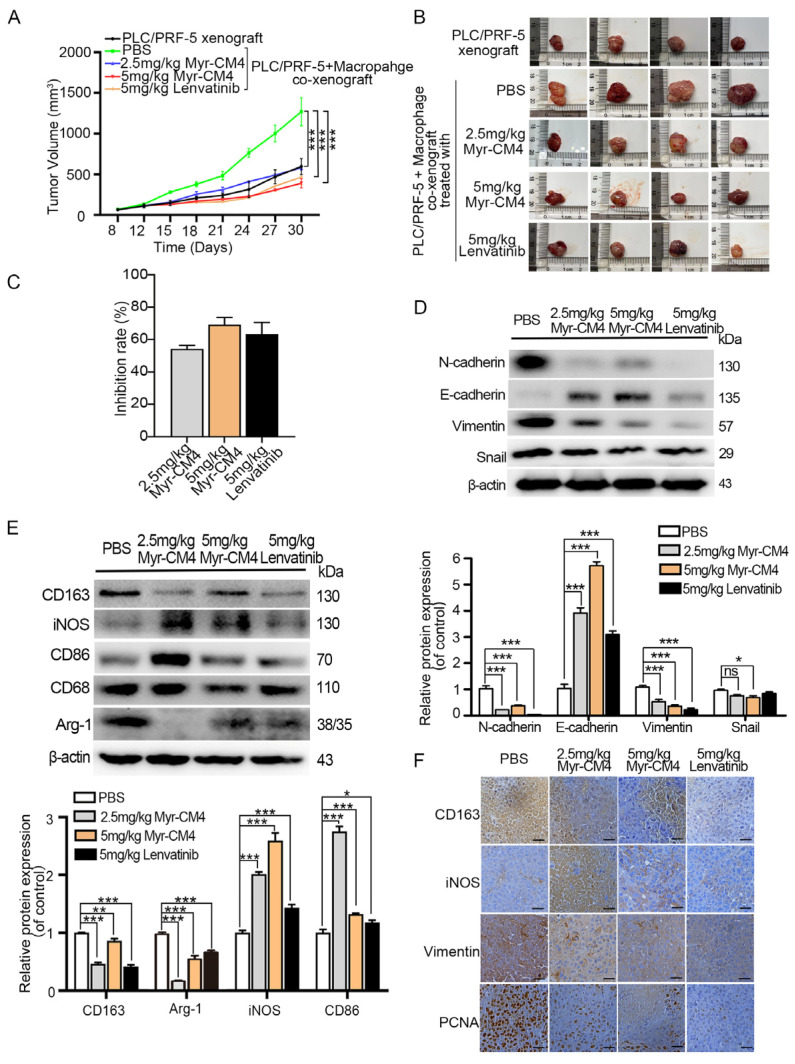
Antitumor effect of myristoyl-CM4 in PLC/PRF-5/macrophage co-xenograft model. PLC/PRF-5 cells and M0 macrophages were injected subcutaneously into nude mice aged 4–6 weeks to generate a co-xenograft mouse models, which were divided into PBS, 2.5 mg/kg myristoyl-CM4, 5 mg/kg myristoyl-CM4, and 5 mg/kg lenvatinib groups. (**A**) Tumor volumes were determined every three days and tumor growth curves over time were graphed. (**B**) Mice were sacrificed on day 30 and the tumors were dissected. (**C**) Comparison of tumor growth inhibition rate among different treatment groups. (**D**) Macrophage, and (**E**) EMT marker levels in tumor tissues were determined using western blotting. (**F**) Paraffin sections were prepared for immunohistochemical staining using anti-iNOS and anti-CD163 to analyze M1 and M2 macrophages. Representative images from each group are shown. “—” represented the scale bar of 100 μm. ns, no significance. * *p* < 0.05, ** *p* < 0.01, *** *p* < 0.001.

## Data Availability

The original contributions presented in this study are included in the article/Appendix A. Further inquiries can be directed to the corresponding author.

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
