# Peer review of "Myristoyl-CM4 Exhibits Direct Anticancer Activity and Immune Modulation in Hepatocellular Carcinoma: Evidence from In Vitro and Mouse Model Studies"

_ijms, 2025, doi:10.3390/ijms26083829_

Round 1
Reviewer 1 Report
Comments and Suggestions for Authors
The authors report that: myristoyl-CM4 exhibits broad-spectrum anticancer activity against several tumor types, maintaining an IC50 in the low micromolar range; and (2) the common mechanism of action involves internalization, targeting mitochondria, and activating mitochondria-dependent apoptosis across different cancer cell types. Mmyristoyl-CM4 exerts effective anti-cancer activity against HCC through both direct and indirect mechanisms. Direct effects on HCC cells include induction of apoptosis and inhibition of EMT, migration, and invasion. Indirectly, myristoyl-CM4 modulates the immune microenvironment by promoting macrophage M1 polarization, contributing to its anticancer effects. These results highlight great potential of myristoyl-CM4 as an effective therapeutic agent for HCC.
Comments
This study is interesting and well written
-i) The dose myristoyl-CM4 effective in this in vitro model il relatively low and therefore it is promising for a possible use in vivo in patients with hepatocellular carcinoma, The same authors have already published studies on the efficacy in vitro of AMPs and in particular Myrystol-CM4i in cell cultures from leukemia and breast cancer
C1 The Authors should highlight in the title and over the manuscript that all the studies were done in cell culture and experimental animals but not in human hepatocarcinoma.
- ii)Line 82: Results A dose-dependent cytotoxic effect was observed following myristoyl-CM4 treatment. The IC50 of myristoyl-CM4 was approximately 4 µM against PLC/PRF-5 82 and HepG2 cells, while the IC50 against LX-2 (a normal human hepatic stellate cell line) was about 18 µM, indicating that myristoyl-CM4 exhibits higher selectivity for HCC cells over normal hepatic cells
C2 Explain whether this dose was used also in animals with cancer. Explain whether the effect on macrophages was only evident with higher concentrations of myristol-CM4, In particular explain whether the anticancer effect has the same mechanism as the anti-inflammatory
-iii)Antimicrobial peptides (AMPs) with high efficiency and low toxicity have demonstrated promising anti cancer properties for cancer treatment. Myristoyl-CM4, a derivative of the CM4 peptide from Bombyx mori, exhibits effective anticancer activity against leukemia and breast cancer cells
C3 Il is important again to specify that no study was performed in human with cancer
-iv) “In this study, we found that at concentrations below the IC50 (1 µM, 2 µM), myristoyl-CM4 effectively inhibited EMT, migration, and invasion in cultured HCC cells”, “Our results showed that myristoyl-CM4 induced apoptosis in HCC cells directly by targeting mitochondria. Additionally, it inhibited the migration and invasion of HCC cells in both monoculture and co-culture systems. Notably, myristoyl-CM4 also promoted M1 macrophage polarization and suppresses M2 polarization in co-culture models both in vitro and in vivo. Furthermore, it demonstrated effective antitumor activity in PLC-PRF-5/macrophage co-xenograft models. These findings highlight the therapeutic potential of myristoyl-CM4 in HCC treatment”.
C4 why these results lead to suggest that the inhibitory effect of myristoyl-CM4 on EMT, migration, and invasion could result from both direct action on HCC cells and indirect modulation through macrophages and which could be the situation in animals in vivo. IT is well known that necrophages and other inflammatory cells (monocytes, dendritic cells) are the mediators of inflammation having both an anti-inflammatory effect but also an inflammatory one
Reviewer 2 Report
Comments and Suggestions for Authors
The paper sounds interesting and can be progressed to the next step based on the editor's decision.
However, the paper should be revised based on the following major comments:
- The abstract section is too long and can be rewritten. The aim of this section is to present the aim of the research, the results, and the main different methods to obtain those results, and all this in 8-13 lines.
-
While the study suggests mitochondrial-dependent apoptosis as the primary mechanism, further mechanistic studies (e.g., RNA sequencing or proteomics) could strengthen the findings.
-
The interaction between Myristoyl-CM4 and mitochondrial proteins should be further explored.
-
Although statistical significance is indicated, additional details on effect sizes and confidence intervals would enhance the robustness of the conclusions.
-
It would be beneficial to include a supplementary table summarizing key statistical analyses.
-
A direct comparison with existing HCC therapies (e.g., tyrosine kinase inhibitors, immunotherapies) would provide context for the potential clinical relevance of Myristoyl-CM4.
-
If available, IC50 values of other AMPs or current therapeutics should be included for comparison.
-
A dedicated section discussing study limitations, such as potential off-target effects or the peptide’s stability in physiological conditions, would improve transparency.
-
Future studies should investigate the long-term effects and pharmacokinetics of Myristoyl-CM4.
- The following papers should be added to the current research:
- 1: Nave, O. (2025). Integral invariant manifold method applied to a mathematical model of osteosarcoma. Results in Control and Optimization, 18(100529), 100529. https://doi.org/10.1016/j.rico.2025.100529
- 2: Li, C., Liu, H., Yang, Y., Xu, X., Lv, T., Zhang, H., Liu, K., Zhang, S., & Chen, Y. (2018). N-myristoylation of Antimicrobial Peptide CM4 Enhances Its Anticancer Activity by Interacting With Cell Membrane and Targeting Mitochondria in Breast Cancer Cells. In Frontiers in Pharmacology (Vol. 9). Frontiers Media SA. https://doi.org/10.3389/fphar.2018.01297
-
Minor grammatical errors and awkward phrasing should be addressed for clarity.
-
The discussion section could be streamlined to avoid redundancy.
Dear editor,
Please see my comments for the authors.
Round 2
Reviewer 2 Report
Comments and Suggestions for Authors
The authors revised the paper based on my major comments.
Comments on the Quality of English LanguagePlease send the paper for professional English editing
Author Response
Comment 1: Please send the paper for professional English editing
Response: In accordance with the journal's requirements, we have submitted our manuscript for professional language editing. Now we provide the final revision and TRACK version, aiming to meet the publication's language standards.